# *In Silico* Insights into HIV-1 Vpu-Tetherin Interactions and Its Mutational Counterparts

**DOI:** 10.3390/medsci7060074

**Published:** 2019-06-22

**Authors:** Patil Sneha, Urmi Shah, Seetharaman Balaji

**Affiliations:** 1School of Biotechnology and Bioinformatics, D Y Patil Deemed to be University, Plot 50, Sector 15, CBD Belapur, Navi Mumbai, Maharashtra 400614, India; sneha.patil@dypatil.edu (P.S.); urmisha.bt16@dypatil.edu (U.S.); 2Research and Development, Bharathiar University, Coimbatore, Tamil Nadu 641046, India; 3Department of Biotechnology, Manipal Institute of Technology, Manipal Academy of Higher Education, Manipal, Karnataka 576104, India

**Keywords:** HIV-1 Vpu, tetherin, transmembrane interactions, aggregation potential

## Abstract

Tetherin, an interferon-induced host protein encoded by the bone marrow stromal antigen 2 (BST2/CD317/HM1.24) gene, is involved in obstructing the release of many retroviruses and other enveloped viruses by cross-linking the budding virus particles to the cell surface. This activity is antagonized in the case of human immunodeficiency virus (HIV)-1 wherein its accessory protein Viral Protein U (Vpu) interacts with tetherin, causing its downregulation from the cell surface. Vpu and tetherin connect through their transmembrane (TM) domains, culminating into events leading to tetherin degradation by recruitment of β-TrCP2. However, mutations in the TM domains of both proteins are reported to act as a resistance mechanism to Vpu countermeasure impacting tetherin’s sensitivity towards Vpu but retaining its antiviral activity. Our study illustrates the binding aspects of blood-derived, brain-derived, and consensus HIV-1 Vpu with tetherin through protein–protein docking. The analysis of the bound complexes confirms the blood-derived Vpu–tetherin complex to have the best binding affinity as compared to other two. The mutations in tetherin and Vpu are devised computationally and are subjected to protein–protein interactions. The complexes are tested for their binding affinities, residue connections, hydrophobic forces, and, finally, the effect of mutation on their interactions. The single point mutations in tetherin at positions L23Y, L24T, and P40T, and triple mutations at {L22S, F44Y, L37I} and {L23T, L37T, T45I}, while single point mutations in Vpu at positions A19H and W23Y and triplet of mutations at {V10K, A11L, A19T}, {V14T, I18T, I26S}, and {A11T, V14L, A15T} have revealed no polar contacts with minimal hydrophobic interactions between Vpu and tetherin, resulting in reduced binding affinity. Additionally, we have explored the aggregation potential of tetherin and its association with the brain-derived Vpu protein. This work is a possible step toward an understanding of Vpu–tetherin interactions.

## 1. Introduction

Tetherin, a protein encoded by the BST2 gene, also known as bone marrow stromal antigen 2/ CD317/HM1.24, is an integral membrane protein involved in the interferon dependent antiviral response pathway [1]. The antiviral activity of tetherin is bestowed upon by its unique topology that blocks the budding viruses and prevents them from leaving the cell [2]. Its N-terminal region is in the cytoplasm with a membrane spanning the helical domain (transmembrane (TM) domain) and an alpha helical coiled coil ectodomain with a glycosyl-phosphatidylinositol modified anchor at C-terminal [3,4]. Tetherin serves as a potent inhibitor of enveloped viruses, like human immunodeficiency virus (HIV), by its tethering phenomena, eventually leading to lysosomal degradation of tethered viral particles [5]. However, HIV-1 establishes a cogent mechanism against this defensive strategy of tetherin by expressing its exclusive accessory protein Viral Protein U (Vpu) [6]. Vpu, an 81-amino acid oligomeric protein, consists of an N-terminal TM domain associated with release of viral particles and a C-terminal domain involved in viral receptors-CD4 degradation [7]. The enhancement of viral budding and release by Vpu is attributed to the dislocation of tetherin from the cell surface causing its internalization and possible lysosomal degradation eventually overcoming its restriction [4]. The phosphorylated conserved serine residues S52 and S56 in the cytoplasmic tail of Vpu are found to be efficient in tetherin degradation [8]. These residues are recognized by an F-box-containing ubiquitin ligase subunit, the beta-transducin repeat-containing protein-2 (β-TrCP-2). Vpu recruits the multi-subunit SCF-β-TrCP E3 ubiquitin ligase complex that causes ubiquitination and degradation of BST-2 [8,9,10]. Thus, its downregulation is partially controlled by β-TrCP, which is also linked to Vpu induced proteosomal CD4 degradation. This is the result of the TM interactions between Vpu and tetherin that aid in bringing the cytoplasmic tail of Vpu in the vicinity of tetherin, leading to its displacement [9]. 

The binding of Vpu to tetherin at their TM domains is a helix–helix interaction that is a crucial for antagonizing antiviral activity [11,12]. The amino acids residing in and around the Vpu–tetherin TM domains present vital aspects in understanding the helix–helix association between the two and determining the susceptibility of tetherin to Vpu. Studies have revealed that the mutations in the TM regions of Vpu or tetherin have rendered tetherin resistant to Vpu antagonism [13,14,15]. A study has shown that a single point mutation in tetherin T45I was successful in rendering tetherin resistant to Vpu-mediated depletion [16], while numerous other studies dealing with identification of interacting points between the two proteins have put forth positions crucial for binding and mutations in them affecting their binding potential [13,14,15,16,17,18,19]. These studies have specified Vpu residues A14, A18, and W22, as well as tetherin residues L22, L23, G25, I26, V30, I33, I34, I36, L37, L41, and T45, to be participating in the interaction [13,14,15,16,17,18,19].

In this study, the *in silico* interactions between HIV-1 Vpu and tetherin were performed. As compartmentalization of HIV-1 in different organs, especially in the central nervous system (CNS), is likely to generate distinct Vpu isolates with varying residues [20,21,22], the Vpu sequences used were isolates of two distinct compartments, the brain and blood. A consensus Vpu sequence was also used in an attempt to highlight the differences in their binding potential. On this basis, the selected amino acid positions of tetherin and blood-derived HIV-1 Vpu were considered for mutational study. The differences in their binding affinities and the interacting residues have been charted out for selected mutations along with the aggregating potential of tetherin.

## 2. Materials and Methods 

### 2.1. Sequence and Structure Retrieval

The representative sequences of blood- and brain-derived HIV-1 Vpu proteins were retrieved from UniprotKB (http://www.uniprot.org/) with accession numbers P35966 and P12516, respectively. These sequences are part of around 50 blood- and 39 brain-derived HIV-1 Vpu protein sequences that were collected and analyzed for their sequence specific variations from different geographical locations [22]. The structure of brain-derived HIV-1 Vpu (P12516) used in the current study has been predicted and validated in our previous work on amyloidogenicity study of HIV-1 Vpu [23]. To analyze the interaction between HIV-1 Vpu and tetherin, the structure of tetherin TM domain (protein data bank (PDB) ID: 2LK9) was retrieved from the protein data bank (http://www.rcsb.org/) [24].

### 2.2. Multiple Sequence Alignment and Generation of Consensus Vpu Sequence

The multiple sequence alignment of geographically divergent 89 blood- and brain-derived HIV-1 Vpu sequences was carried out in Clustal Omega (https://www.ebi.ac.uk/Tools/msa/clustalo/). Clustal Omega applies seeded guide trees and Hidden Markov Model (HMM) profile–profile methods for ensuring an optimal alignment between the given sequences [25]. The consensus sequence was obtained from an Emboss explorer, a server for creating consensus sequence from multiple alignment (http://www.bioinformatics.nl/cgi-bin/emboss/cons) [26]. The consensus was deduced with a default plurality value taken as half the total weight of all the sequences in the alignment. The variations in blood and brain Vpu residue positions from consensus Vpu sequence generated are represented in Figure 1. The geographically and compartmentally distinct HIV-1 Vpu proteins were compiled together in a consensus sequence to form a representative of a complete Vpu blood and brain dataset and further aid in understanding the interactions between Vpu–tetherin complexes.

### 2.3. Protein Structure Modeling and Validation

The tertiary structures of representative blood-derived Vpu and consensus Vpu sequences were modeled based on homology. A BLAST similarity search [27] was performed against the PDB database with a Blosum62 substitution matrix and default parameters to select a template with a good alignment score and maximum query coverage. The template with PDB ID: 2N28, having an identity score of 75% and query coverage of 72% for blood-derived Vpu sequence and identity score of 82% and query coverage of 73% for consensus Vpu sequence, was selected for homology modeling. The structure of the selected template (PDB ID: 2N28) determined using the Nuclear Magnetic Resonance (NMR) method was retrieved from PDB (http://www.rcsb.org/) [24]. The Vpu structures were modeled using the SWISS-MODEL server, an automated protein homology-modeling server (https://swissmodel.expasy.org/) [28]. Energy minimization was done using the inbuilt GROMOS96 force field. The overall model quality estimation and validation of blood- and brain-derived Vpu structures was done in protein structure analysis (ProSA-web) tool (https://prosa.services.came.sbg.ac.at/prosa.php) and the quality was assessed based on Ramachandran’s plot using PROCHECK (http://servicesn.mbi.ucla.edu/PROCHECK/) [29,30].

### 2.4. Protein–Protein Interaction 

In order to comprehend the binding profile of Vpu with tetherin, the wild-type representative blood- and brain-derived Vpu structures and consensus Vpu structure were subjected for interactions with the TM tetherin structure (ID: 2LK9). This was performed using Hex version 8.0.0, an interactive molecular graphics application for performing molecular interactions. Hex enables modeling of each molecule, employing 3D extensions of real orthogonal spherical polar built functions encrypting both surface shape and electrostatic charge and potential distributions. Hex illustrates the surface profiles of proteins applying a two-term surface skin and Van der Waals steric density model. With the use of appropriate scaling factors, the docking score is inferred as a minimized interaction energy [31,32]. Docking was performed using a reference complex describing the anti-parallel orientation of Vpu and tetherin, as reported in the available literature [11]. The set of interacting complexes were then submitted to PROtein binDIng enerGY prediction (PRODIGY) web server for prediction of binding affinity in protein–protein complexes based on intermolecular interaction contacts and characteristics resulting from non-interface surface. (https://nestor.science.uu.nl/prodigy/) [33,34]. The complexes were visualized and analyzed in PyMOL v 2.2.3 [35] and Swiss PDB Viewer (SPDBV) version 4.10 or “Deep view” [36]. The interacting residues between Vpu and tetherin were scrutinized in a proteins interaction calculator (PIC) server that evaluates various hydrophobic and ionic interactions, hydrogen bonds, disulfide bridges, and aromatics interactions between the proteins comprising the complexes (http://pic.mbu.iisc.ernet.in/) [37].

### 2.5. Mutational Study

The possible mutations were analyzed in I-Mutant 3.0 [38] and Sorting Intolerant From Tolerant, SIFT [39]. I-Mutant3.0 is a Support Vector Machine-based web server for predicting the effect of single point mutations on protein stability taking protein sequence or structure as input. SIFT predicts whether an amino acid substitution disturbs the protein function. SIFT prediction is grounded on the degree of conservation of amino acid residues in sequence alignments resulting from closely related sequences, composed through PSI-BLAST. The mutations in the protein structures are then performed in Chimera v 1.11.2, a protein visualization and analysis tool [40]. The structure editing menu option in Chimera aids in selection of best probability rotamer for the defined mutation at the desired position. The following amino acids in tetherin, such as L22, L23, L24, G25, I26, L29, V30, I33, I34, I36, L37, V39, P40, L41, F44, and T45, were selected for mutations (Table 1). Similarly, the following amino acids of blood-derived Vpu, such as S3, Q5, L7, A8, A11, V14, A15, I17, A19, W23, I25, F27, R31, and K32, were selected for mutations (Table 2). Mutations were introduced in the blood-derived Vpu, as it had the best binding affinity of −5.0 kcal/mol(ΔG) in Vpu–tetherin docked complexes in comparison with the brain-derived and consensus Vpu structures. Each selected position in tetherin and Vpu were mutated with synonymous, as well as non-synonymous, substitutions and tested for tolerance and protein stability in I-Mutant 3.0 and SIFT (Table 1 and Table 2). The reliability index (RI) is computed only when the sign of the stability change is predicted and DDG (kJ/mol) indicates the free energy change upon mutation. SIFT score calculates the effect of amino acid substitution on protein function ranging from 0.0 (deleterious) to 1.0 (tolerated).

### 2.6. Aggregation Potential Prediction of Tetherin

There are several online tools and web servers available to estimate amyloidogenic regions from protein sequences based on complex biological mechanisms linked to amyloidosis and are based on consider diverse physicochemical features, such as charge, secondary-structure propensity, and hydrophobicity [41,42]. Formation of cross β-sheet arrangement is a principle characteristic of an amyloid and has a prime contribution in identification of potentially aggregating regions [43]. The amyloidogenic region prediction of tetherin was done in Fold Amyloid [44], AGGRESCAN [45], TANGO [46], WALTZ [47], MetAmyl [48], AMYLPRED2 [49], and PASTA [50] programs that differ in their algorithms, and each has own applications for predictions of aggregation-prone sites or specific segments of proteins that can tend to aggregate. These tools integrate diverse properties of proteins accountable for amyloidogenicity, such as hydrophobicity, aggregation propensity scale, β-strand contiguity, average packing density, hexapeptide conformational energy, and possible conformational switches between α-helix and β-sheet [51,52,53]. The secondary structure prediction for tetherin is performed in the Chou–Fasman server (http://cho-fas.sourceforge.net/) to derive the relative frequencies of each amino acid in α-helices, β-sheets, and coils based on known protein crystal structures solved with X-ray crystallography [54]. The TANGO server predicts aggregation nucleating regions in proteins, as well as the effect of mutations and environmental conditions on the aggregation propensity of these regions [46]. Identification of discordant region in tetherin was performed by estimating the amino acid propensity values from the TANGO (http://tango.crg.es/n) web server. It provides propensities for each amino acid in the sequence to form either α-helix or β-sheet. The possible conformational switches in the query sequence were predicted based on secondary structure. Conformational switches are regions that can alter their shape in response to certain input signals either ligand binding, chemical binding, or environmental conditions [55]. The amino acids in such stretches have propensities to form both helix and sheet encompassing a α-helix/β-sheet discordant region.

## 3. Results

### 3.1. Modeled Protein Structures and Evaluation

The 3D structures of blood- and brain-derived Vpu and consensus Vpu were modeled using the SWISS-MODEL server and assessed the overall quality in PROSA giving a z-score of 0.52 for Vpu from blood isolate, −0.14 for Vpu from brain isolate, and 0.52 for the consensus Vpu structure. The z-score for all the modeled Vpu structures were found to fall in the range of scores typically found for native proteins of similar size. The Ramachandran plot statistics of the modeled Vpu structures presented in Table 3 had over 90% of residues in the most favorable region, indicating good overall model quality. The minimized energy as calculated in Swiss PDB Viewer (SPDBV) was about −592.97 kcal/mol for blood-derived Vpu; for brain-derived Vpu, it was −615.91 kcal/mol, and, for the consensus Vpu structure, it was −763.86 kcal/mol.

### 3.2. Docking Studies of Wild-Type Structures

Protein–protein interactions performed for the modeled Vpu structures with tetherin in Hex provided around 30–50 possible docked confirmations that were then manually visualized for their anti-parallel orientation and interacting positions and were tested in the PIC server for estimating the interacting residues. Our focus had been mainly on deriving the maximum interacting residues between the TM regions of both proteins Vpu and tetherin, as it has been reported that the antagonism of Vpu to the anti-viral activity of tetherin is by intermolecular interactions between the helix–helix TM domains of both the proteins [12,14]. Thus, such complexes were manually selected and submitted to PRODIGY for evaluating the finest docked confirmation with maximum interacting residues and best binding affinity. The value of the dissociation constant (Kd) is calculated at 37 °C in PRODIGY. The selected complexes, their binding affinities, and Kd values are represented in Table 4.

The residue pairs forming hydrophobic interactions within 5 Å in wild-type blood-derived Vpu–tetherin complex having a binding affinity (ΔG) of −5.0 kcal/mol and Kd of 6.4 × 10^−4^ M were obtained for A8-I42, A8-V39, I9-I42, I9-L41, I9-P40, I9-V39, V10-I43, V10-I42, V10-P40, V10-V39, A11-V39, L12-V39, L12-I36, L12-V35, V13-P40, V13-V39, V13-L37, V13-I36, V14-I36, A15-I36, I17-I36, A19-L32, I20-I33, I20-L32, I20-L29, W23-L29, W23-I28, I25-L22, F27-I28, F27-I26, F27-L24, F27-L23, F27-L22, I28-I26, and I28-L22, as presented in Figure 2. The anti-parallel interactions between the TM domains of blood derived Vpu and tetherin is are presented in Figure 3. The complex had a potential energy of −314.17 kcal/mol as computed by force field parameters. The residue pairs forming hydrophobic interactions within 5 Å in wild-type brain-derived Vpu–tetherin complex having a binding affinity (ΔG) of −3.8 kcal/mol and Kd of 2.0 × 10^−3^ M were obtained for V10-P40, V10-V39, V10-L37, V10-I36, A11-L37, V13-I36, V14-L37, V14-I36, V14-I34, V14-I33, I17-I36, I17-I33, I17-L32, I17-L29, I18-I33, I18-V30, I18-L29, I20-L29, V21-V30, V21-L29, V21-I28, V21-I26, I25-L23, I25-L22, and I39-L22, as presented in Figure 2. The anti-parallel interactions between the TM domains of brain derived Vpu and tetherin are presented in Figure 4. A potential energy computed by force field parameters was -148.65 kcal/mol for this complex. The residue pairs forming hydrophobic interactions within 5 Å in wild-type consensus Vpu–tetherin complex having a binding affinity (ΔG) of −4.3 kcal/mol and Kd of 9.8 × 10^−4^ M were obtained for L7-F44, L7-I43, L7-L41, A8-I43, A8-P40, V10-P40, V10-L37, A11-L37, L12-L37, V14-L37, V14-I36, V14-I34, V14-I33, A15-L37, I17-I33, I17-L29, I18-I34, I18-I33, I18-V30, V21-V30, V21-L29, V21-I26, V22-V30, I25-I26, I25-L24, I25-L23, V26-L23, and I28-L23, as presented in Figure 2. The complex had a potential energy of −261.08 kcal/mol, as computed by force field parameters. The anti-parallel interaction between the TM domains of consensus Vpu and tetherin are presented in Figure 5. There were no direct bonds observed in the docked complexes.

### 3.3. Docking Studies of Mutant Structures

The amino acids of tetherin at positions L22, L23, L24, G25, I26, L29, V30, I33, I34, I36, L37, V39, P40, L41, F44, and T45 and those of blood-derived Vpu at positions S3, Q5, L7, A8, A11, V14, A15, I17, A19, W23, F27, R31, and K32 were subjected to synonymous and non-synonymous mutations, and their results for stability and tolerance is given in Table 1 and Table 2. Based on these scores, few mutations were shortlisted, and tetherin and Vpu structures were mutated in Chimera. The protein–protein interactions were performed and analyzed for these mutant structures. The selected mutations of Vpu and tetherin, interacting residues between them, and binding energy of the complexes are listed in Table 5 and Table 6. All the mutated structures of tetherin and Vpu exhibited a decrease in binding energies in comparison with the wild-type blood-derived Vpu–tetherin complex. The single and triple mutations that resulted in reduced binding affinities and fewer hydrophobic connections as compared to other mutations are represented in Figure 6 and Figure 7. It was observed that the following set of residues, A11, V13, V14, A15, I17, I25, and I28, of Vpu and L22, L23, L24, I33, V30, I36, L37, P40, I43, and F44 of tetherin were found interacting in most of the mutant structures, indicating their critical role in the formation of binding complexes that remain unaffected despite of mutations. 

### 3.4. Amyloidogenicity Prediction of Tetherin

The aggregation prediction results of tetherin showed the presence of amyloid forming regions at positions ranging from 21 to 47, as well as from 161 to 176 derived out of consensus output from at least six out of seven servers (Table 7). The region 21–47 forms the TM region that is known to interact with Vpu [56]. The various amyloid prediction servers employ different algorithms based on aggregation propensity scale, expected packing density, physicochemical properties of secondary structure, βstrand contiguity, and formation of possible conformational switches and presents the potential aggregating regions as hotspots, as represented in Table 7. Parallel aggregation within the predicted region of 21–47 is observed in PASTA 2.0 with energy of −20.54 PASTA Energy Units (1 PEU = 1.192 kcal/mol), as shown in Figure 8A. The program predicts region in query sequence likely to form β-strand inter-molecular pairing, thus identifying the aggregation probability of the region [50]. The aggregation energy indicates a low energy cross β-structure conformation of predicted stretch signifying a stabilized assembly, while the other predicted stretch of 161–176 comprises a disordered region, as indicated in Figure 8B, presenting aggregation and disorder profile of tetherin. Discordant regions predicted by secondary structure in tetherein lies at positions 28–34, 42–46, 68–69, 93–96, 144–148, and 167–175. The propensity values for tetherin as predicted in Chou–Fasman for identified aggregating region 21–47 were 96.3% for helix and 88.9% for sheet and for region 161–176 were 75% for helix and 37.5% for sheet (Table 8). Region 21–47 of tetherin does show the presence of amino acids with high propensity for both helix and sheet. The aggregating regions in tetherin are predicted at positions 22–38 and 168–179. The helix and sheet aggregation plots are presented in Figure 9.

## 4. Discussion

Tetherin, a type II membrane protein, blocks the release of variety of enveloped viruses by retaining them on the cell surfaces, whereas HIV-1 is known to successfully overcome this blocking by crosslinking its Vpu protein with tetherin, making the virus resistant to the anti-viral defense mechanism [1,2]. Vpu directly interacts with the tetherin TM domain through its AxxxAxxxA motif present in the TM domain [12]. It is an established fact that HIV-1 Vpu is a highly variable protein; however, factors that contribute to Vpu sequence variability are not well defined [21]. 

The host immune responses may be attributed to the variability wherein polymorphism is acquired as a means of immune escape or other functional benefit [21]. Additionally, compartmentalization of HIV-1 does serve as an important aspect that would drive the sequence diversity, likely due to differential immune selection pressures, cell type-specific differences in gene expressions, and local concentrations of antiviral drugs or co-infections altering the microenvironment [20]. Compartmentalization of HIV-1 in the CNS has been very well documented and massively studied to probe the mechanisms leading to pathogenesis in HIV associated dementia (HAD) [57,58]. Case in point, the Vpu–tetherin interactions studied here have considered the HIV-1 Vpu sequences from both blood and brain compartments in a view to understand the consequence of sequence variability on binding profiles between Vpu–tetherin complexes. Our study represents the exact residues forming hydrophobic connections between HIV-1 Vpu from blood- and brain-derived isolates, and with tetherin, demonstrates the differences in the binding residues and their affinities. The consensus sequence derived from blood- and brain-derived Vpu sequences is also taken into account for comparison of the variations in interacting residues. Protein–protein interaction data depicts the blood-derived isolate to have maximum binding affinity with tetherin having a ΔG value of −5.0 kcal/mol, while for the brain-derived and consensus Vpu sequences, the binding affinity decreases to −3.8 and −4.3 kcal/mol, respectively. The N-terminal region of Vpu that interacts with tetherin is noted to encompass a region with α-helix/β-sheet discordance and the same was reflected in 70ns simulation performed on brain-derived Vpu structure [23]. However, the blood-derived Vpu structure failed to show such a transition in simulations highlighting the differential behavior of Vpu from blood and brain compartments suggestively attributed to variations in the sequence.

Specific mutations in the TM region of both Vpu or tetherin are demonstrated to be essential in rendering tetherin resistant to the Vpu counteraction [16]. From a wide range of mutations introduced in blood-derived Vpu, as well as the tetherin TM domain, the interacting domains were verified by protein–protein docking (as indicated in Table 5 and Table 6). Our study puts forth few single point as well as multiple (triple) mutations that effectively resulted in minimal hydrophobic interactions and reduced binding affinity. About 21 mutations performed in tetherin at positions L22, L23, L24, G25, I26, L29, V30, I33, I34, I36, L37, V39, P40, L41, F44, and T45, wherein single point mutations of L23→Y, L24→T, and P40→T depicted a decrease in residue pairs involved in hydrophobic interactions in the Vpu-mutant tetherin complex. Hydrophobic contacts were observed between 4 to 6 Vpu and tetherin residues, as represented in Figure 6. The binding affinities (ΔG) of the L23Y, L24T, and P40T mutants were −2.6, −2.2, and −2.7 kcal/mol with corresponding dissociation constants (Kd) of 1.2 × 10^−3^, 2.9 × 10^−3^, and 1.9 × 10^−3^ M, respectively. The rest of the mutations showed a reduction in binding affinity between Vpu and mutant tetherin molecules. There were about 44 possible complexes with all possible conformations between the two proteins, indicating at least few interacting residues in each conformation. The reported T45I single mutation that causes tetherin to become resistant to Vpu-mediated depletion [16] was also tested in our study. However, the protein–protein complex was found to be associated with few hydrophobic residue interactions and binding affinity of −2.5 kcal/mol. The combined/multiple mutations at {L22S, F44Y, L37I} and {L23T, L37T, T45I} showed minimal hydrophobic connections and low binding affinities in Vpu–tetherin complexes with binding affinities (ΔG) and dissociation constant (Kd) of −2.2 kcal/mol and 1.8 × 10^−2^ M and −2.9 kcal/mol and 2.3 × 10^−2^ M, respectively. Interestingly, these sets of residues are observed to be interacting in most of the mutant tetherin-Vpu complexes. A computer-assisted structural modeling and mutagenesis study predicted that an alignment of four amino acid residues such as I34, L37, L41, and T45 on the same helical face in the TM domain of tetherin is important for the Vpu-mediated antagonism of human tetherin [14]. Hence, the mutations in these residues may prove crucial in breaking the contacts and decreasing the binding affinity. Out of the 22 mutations performed in blood-derived Vpu at S3, Q5, L7, A8, V10, A11, V14, A15, I17, I18, A19, W23, I25, F27, R31, and K32 single point mutations of A19→H and W23→Y affected the hydrophobic interactions in mutant Vpu and tetherin, while combined mutations of {V10K, A11L, A19T}, {V14T, I18T, I26S}, and {A11T, V14L, A15T} showed minimal hydrophobic connections and low binding affinities in the docked complexes, as represented in Figure 7. The docked complexes of A19H and W23Y Vpu mutants possessed binding affinities (ΔG) as −2.5 and −2.1 kcal/mol and Kd of 2.9 × 10^−3^ and 2.1 × 10^−3^ M, while the complexes with triple mutations {V10K, A11L, A19T}, {V14T, I18T, I26S}, and {A11T, V14L, A15T} had binding affinities (ΔG) of −2.6, −2.7, and −2.3 kcal/mol and the corresponding Kds of 2.1 × 10^−5^, 2.2 × 10^−4^, and 2.8 × 10^−2^ M, respectively. The amino acids A11, V13, V14, A15, I17, I25, and I28 of Vpu are frequently observed in the interactions of mutant Vpu–tetherin complexes. A study associating the biological activity of the TM hetero-dimers of HIV-1 Vpu and its host factors with computationally-derived structural features suggests the importance of the tilt in Vpu’s alanine rim Ala-8/11/15/19 [17]. This study implicates the reduction in parallel alignment to correlate with low activity. On similar lines, these alanine residues are mutated in our study and, as mutation in A15F is observed to have higher downregulation activity than A19F in a study by Li et al., A15T and A19H mutation in our study resulted in reduced hydrophobic interactions, suggesting the absence of a strong connection must be impacting the tilt of the Alanine rim. In a futuristic view, tetherin or Vpu binding agents might prove essential in protecting tetherin from viral encoded counteractions and enhance its anti-viral properties. Interestingly, a study has also reported that a cholesterol-binding compound, Amphotericin B methyl ester, inhibits the HIV-1 assembly and releases by interfering with the anti-CD317/BST-2/tetherin function of Vpu [59].

Another aspect of this study was the investigation of aggregating potential of tetherin. The various amyloid predicting programs suggested the presence of amyloidogenic stretch at position 21–47 and 161–176 in tetherin. The region 21–47 is a TM helical region which is also predicted to encompass a α-helix/β-sheet discordant region with residues having higher propensities for helix, as well as sheet. It is a well-known fact that the neurodegenerative diseases are characterized by protein aggregation and deposition of insoluble amyloid fibrils in the brain [60]. Also, a presence of discordant region is quite a common property in these amyloidogenic proteins [61]. Moreover, these features are also shared in the development of HAD/HIV-associated neurocognitive disorders (HAND) [62,63].

Numerous studies have presented findings of amyloid aggregates in the brains of HIV-infected patients, either due to accumulation of β-amyloid precursors or viral-derived aggregating proteins [64,65,66,67,68,69]. However, work on recognizing the exact mechanisms leading to pathogenesis in HAD and role of amyloids in its enhancement is still underway. Functionally, HIV-1 Vpu is involved in directing the ubiquitination and degradation of tetherin by interacting with its TM region, wherein Vpu acts as an adapter molecule linking tetherin to the cellular ubiquitination machinery via βTrCP [10]. It is prospective that, during this degradation process, tetherin could possibly aggregate and evade the activity of ubiquitin-proteasome machinery. Although almost all of the proteins encoded by the human genome can be efficiently removed from the cell when misfolded, a number of polypeptides produced from post-translational conjugation, such as hyperphosphorylated tau in Alzheimer’s disease (AD), or due to endoproteolytic cleavage, such as amyloid β peptides, tend to be rapidly aggregated into oligomers enriched in β-sheet and escape the regular degradation process [70,71,72,73]. Such oligomers are at least partially resistant to all known proteolytic pathways, further leading to inclusion bodies or extracellular plaques having highly ordered β-sheet fibrils [70]. On these lines, the predicted amyloidogenic potential of tetherin needs to be further studied to affirm this possibility and investigate its involvement in progress of dementia.

## 5. Conclusions

The observation of best binding affinity between blood-derived Vpu and tetherin, in comparison to brain-derived and consensus HIV-1 Vpu proteins, reflects the effect of sequence variations in compartmentalized isolates on their binding potential to tetherin. The computational protein–protein interaction of tetherin and Vpu mutant complexes highlights the consistent hydrophobic interaction of key residues A11, V13, V14, A15, I17, I25, and I28 of Vpu and L22, L23, L24, I33, V30, I36, L37, P40, I43, and F44 of tetherin in the majority of the complexes, despite the various mutations, suggesting their essential involvement in binding. The extensive mutational analysis further charts out that the selective single point and/or triple mutations in the residues at positions L22, L23, L24, L37, P40, and F44 of tetherin and V10, A11, V14, A15, I18, A19, W23, and I26 of Vpu results in a decrease in hydrophobic interactions and reduced binding affinities. Additionally, amyloidogenecity prediction of tetherin has revealed its possible aggregation potential that needs to be further explored for its underlying contribution in dementia progression. Our study provides a basic approach to investigate the interacting residues and possible mutations, as well as to understand the connectivity aspects between HIV-1 Vpu and tetherin. 

## Figures and Tables

**Figure 1 medsci-07-00074-f001:**
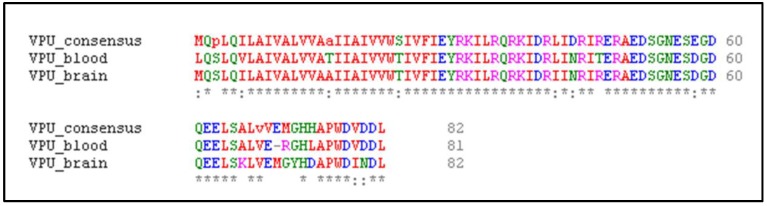
Representation of identity (*) and conserved substitutions (:) between human immunodeficiency virus (HIV)-1 Viral Protein U (Vpu) sequences from blood and brain isolates and the consensus Vpu sequence.

**Figure 2 medsci-07-00074-f002:**
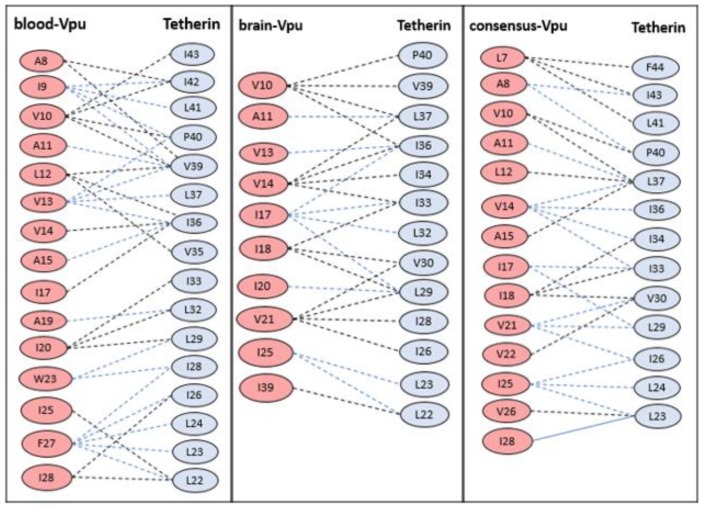
The hydrophobic interactions within 5 Å for wild-type blood-derived Vpu–tetherin complex, wild-type brain-derived Vpu–tetherin complex, and wild-type consensus Vpu–tetherin complex.

**Figure 3 medsci-07-00074-f003:**
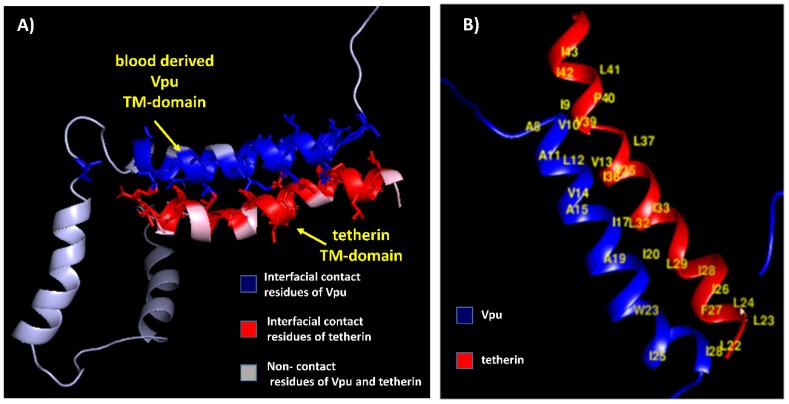
(**A**) Blood-derived Vpu–tetherin complex with interfacial contact residues of Vpu in blue, interfacial residues of tetherin in red, and non-contact residues in grey. (**B**) Residues forming hydrophobic interactions (labeled in yellow) between blood-derived Vpu and tetherin.

**Figure 4 medsci-07-00074-f004:**
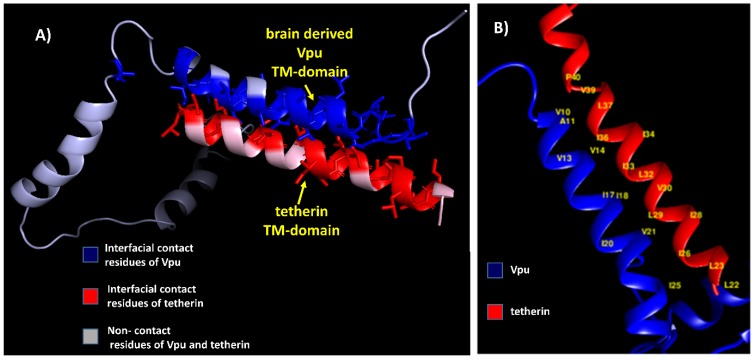
(**A**) Brain-derived Vpu–tetherin complex with interfacial contact residues of Vpu in blue, interfacial residues of tetherin in red, and non-contact residues in grey. (**B**) Residues forming hydrophobic interactions (labeled in yellow) between brain-derived Vpu and tetherin.

**Figure 5 medsci-07-00074-f005:**
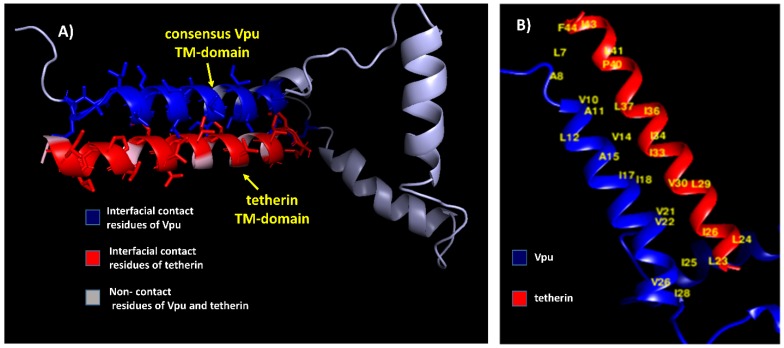
(**A**) Consensus Vpu–tetherin complex with interfacial contact residues of Vpu in blue, interfacial residues of tetherin red, and non-contact residues in grey. (**B**) Residues forming hydrophobic interactions (labeled in yellow) between consensus Vpu and tetherin.

**Figure 6 medsci-07-00074-f006:**
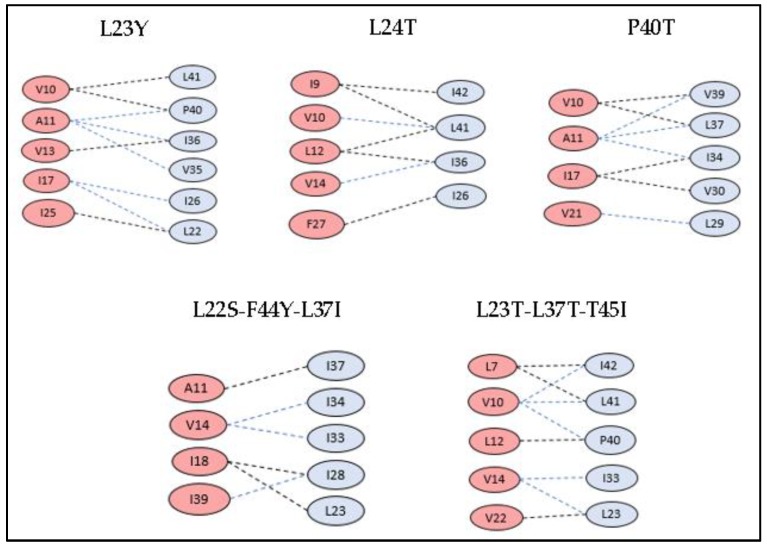
The hydrophobic interactions within 5 Å for tetherin mutants having reduced binding affinities and minimal hydrophobic connections.

**Figure 7 medsci-07-00074-f007:**
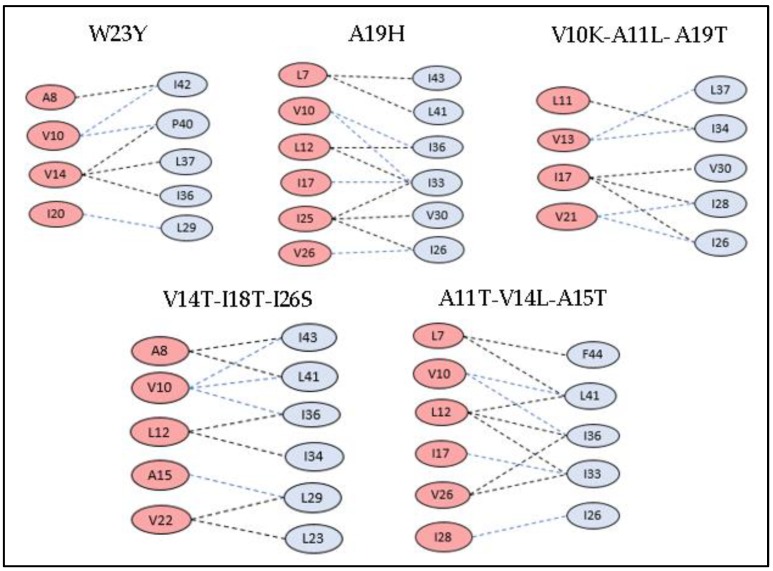
The hydrophobic interactions within 5 Å for Vpu mutants having reduced binding affinities and minimal hydrophobic connections.

**Figure 8 medsci-07-00074-f008:**
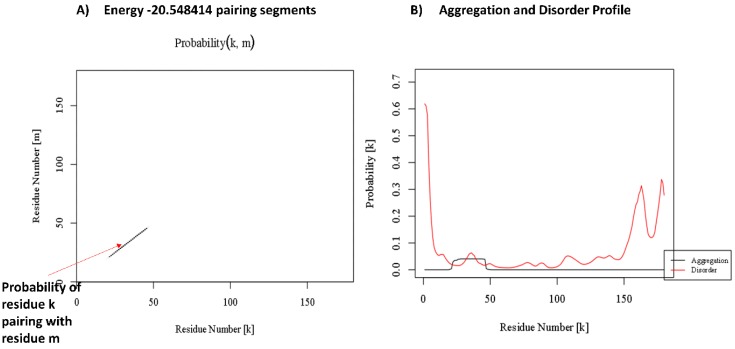
(**A**) Pairing results in PASTA2.0 for tetherin. (**B**) Aggregation and disorder profile of tetherin-derived in PASTA2.0.

**Figure 9 medsci-07-00074-f009:**
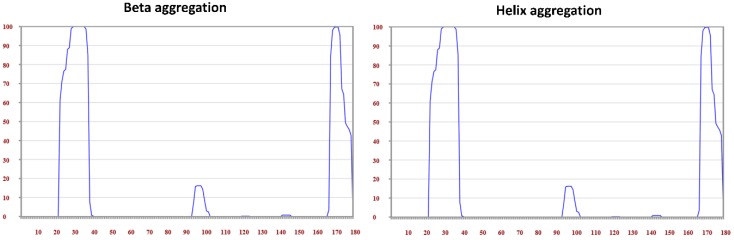
Beta aggregation and Helix aggregation plots generated in TANGO highlighting regions 22—38 and 168—179 to have high aggregation probability.

**Table 1 medsci-07-00074-t001:** Mutation Analysis of tetherin in I-Mutant 3.0 and Sorting Intolerant From Tolerant (SIFT). RI = reliability index.

Position	Substitution	Characteristic	SIFT Tolerated	SIFT Score	I-Mutant Stability	RI	DDG (kJ/mol)
22	L → S	Uncharged Polar	No	0.01	Decrease	8	−1.31
	L → M	Nonpolar	Yes	0.07	Decrease	5	−0.85
	L → K	Basic	No	0.01	Decrease	8	−1.36
	L → F	Nonpolar	Yes	0.05	Decrease	7	−0.89
	L →Y	Uncharged Polar	No	0.01	Decrease	2	−1.07
23	L →T	Uncharged Polar	Yes	0.07	Decrease	8	−1.52
	L → S	Uncharged Polar	No	0.04	Decrease	9	−1.55
	L → H	Basic	No	0.03	Decrease	9	−1.72
	L → V	Nonpolar	Yes	0.43	Decrease	6	−0.91
	L → R	Basic	Yes	0.11	Decrease	5	−1.15
	L → Y	Uncharged Polar	No	0.02	Decrease	2	−1.07
24	L → T	Uncharged Polar	No	0.04	Decrease	8	−1.56
	L → R	Basic	No	0.01	Decrease	5	−1.19
	L →M	Nonpolar	Yes	0.09	Decrease	6	−0.97
	L→ I	Nonpolar	Yes	0.63	Decrease	7	−1.10
	L → E	Acidic	No	0.01	Decrease	6	−1.41
25	G → A	Nonpolar	Yes	1.00	Decrease	1	−0.57
	G → C	Nonpolar	Yes	0.09	Decrease	6	−0.98
	G → T	Uncharged Polar	Yes	0.16	Decrease	7	−0.79
	G → L	Nonpolar	Yes	0.10	Decrease	6	−0.53
	G → Y	Uncharged Polar	No	0.03	Decrease	3	−0.79
26	I → S	Uncharged Polar	Yes	0.05	Decrease	9	−1.33
	I → L	Nonpolar	Yes	0.34	Decrease	8	−0.63
	I → D	Acidic	No	0.02	Decrease	8	−1.29
	I → N	Uncharged Polar	No	0.03	Decrease	8	−1.26
29	L → D	Acidic	No	0.04	Decrease	8	−1.82
	L → F	Nonpolar	Yes	0.13	Decrease	8	−1.20
	L → K	Basic	Yes	0.08	Decrease	9	−1.88
	L → V	Nonpolar	Yes	0.28	Decrease	7	−1.14
30	V → G	Nonpolar	Yes	0.55	Decrease	10	−2.06
	V → H	Basic	No	0.03	Decrease	10	−1.87
	V → E	Acidic	Yes	0.10	Decrease	8	−1.54
	V → Q	Uncharged Polar	Yes	0.07	Decrease	9	−1.52
	V → A	Nonpolar	Yes	1.00	Decrease	9	−1.29
33	I → T	Uncharged Polar	No	0.01	Decrease	9	−1.46
	I → F	Nonpolar	Yes	0.09	Decrease	9	−1.00
	I → K	Basic	No	0.00	Decrease	9	−1.59
	I → V	Nonpolar	Yes	1.00	Decrease	6	−0.45
34	I → T	Uncharged Polar	No	0.04	Decrease	9	−1.50
	I → G	Uncharged Polar	Yes	0.09	Decrease	9	−2.09
	I → F	Nonpolar	Yes	0.07	Decrease	9	−0.98
	I → L	Nonpolar	Yes	0.45	Decrease	8	−0.83
36	I → G	Uncharged Polar	Yes	0.17	Decrease	9	−2.05
	I → A	Nonpolar	Yes	0.38	Decrease	9	−1.71
	I → S	Uncharged Polar	Yes	0.11	Decrease	9	−1.61
	I → F	Nonpolar	Yes	0.14	Decrease	8	−0.96
	I → K	Basic	Yes	0.07	Decrease	9	−1.64
37	L → T	Uncharged Polar	No	0.00	Decrease	9	−1.86
	L → M	Nonpolar	No	0.05	Decrease	8	−1.13
	L → I	Nonpolar	No	0.04	Decrease	9	−1.30
	L → V	Nonpolar	Yes	0.30	Decrease	8	−1.13
39	V → A	Nonpolar	Yes	0.15	Decrease	9	−1.24
	V → D	Acidic	No	0.01	Decrease	9	−1.57
	V → K	Basic	No	0.03	Decrease	10	−1.75
	V → T	Uncharged Polar	Yes	0.12	Decrease	10	−1.26
40	P → T	Uncharged Polar	No	0.05	Decrease	8	−0.99
	P → A	Nonpolar	Yes	0.09	Decrease	8	−1.10
	P → D	Acidic	No	0.01	Decrease	8	−1.23
	P → N	Uncharged Polar	No	0.02	Decrease	8	−1.38
	P → F	Nonpolar	Yes	0.19	Decrease	8	−0.80
41	L → Y	Uncharged Polar	No	0.00	Decrease	6	−1.28
	L → F	Nonpolar	Yes	0.06	Decrease	8	−1.10
	L → A	Nonpolar	Yes	0.06	Decrease	9	−1.75
	L → T	Uncharged Polar	Yes	0.09	Decrease	8	−1.57
44	F → S	Uncharged Polar	No	0.03	Decrease	8	−1.15
	F → Y	Uncharged Polar	Yes	1.00	Decrease	2	−0.74
	F → I	Nonpolar	Yes	0.19	Decrease	5	−0.80
45	T → I	Nonpolar	Yes	0.62	Decrease	6	−0.77
	T → N	Uncharged Polar	No	0.03	Increase	0	−0.56
	T → L	Nonpolar	Yes	0.28	Decrease	7	−0.70
	T → Y	Uncharged Polar	No	0.01	Decrease	3	−0.51

**Table 2 medsci-07-00074-t002:** Mutation Analysis of Vpu in I-Mutant 3.0 and SIFT.

Position	Substitution	Characteristic	SIFT Tolerated	SIFT Score	I-Mutant Stability	RI	DDG (kJ/mol)
3	S → T	Uncharged Polar	Yes	0.40	Increase	1	−0.05
	S → Y	Uncharged Polar	Yes	0.27	Increase	4	−0.23
	S → D	Acidic	Yes	0.49	Increase	6	0.06
	S → M	Basic	No	0.04	Increase	4	0.00
5	Q → V	Nonpolar	Yes	0.33	Increase	0	0.11
	Q → E	Acidic	Yes	1.00	Increase	3	−0.40
	Q → L	Nonpolar	Yes	0.40	Increase	2	0.01
	Q → G	Nonpolar	Yes	0.37	Decrease	7	−0.80
7	L → S	Uncharged Polar	Yes	0.06	Decrease	9	−1.38
	L → Y	Uncharged Polar	No	0.01	Decrease	4	−1.03
	L → K	Basic	No	0.01	Decrease	8	−1.30
	L → V	Nonpolar	Yes	0.23	Decrease	6	−0.72
	L → I	Nonpolar	Yes	0.52	Decrease	8	−0.94
8	A → T	Uncharged Polar	No	0.04	Decrease	9	−1.22
	A → G	Nonpolar	Yes	0.93	Decrease	9	−1.62
	A → S	Uncharged Polar	Yes	0.20	Decrease	10	−1.32
	A → N	Uncharged Polar	No	0.02	Decrease	9	−1.21
11	A → F	Nonpolar	No	0.00	Decrease	9	−0.77
	A → S	Uncharged Polar	Yes	0.09	Decrease	10	−1.18
	A →G	Nonpolar	Yes	0.14	Decrease	10	−1.51
	A → E	Acidic	Yes	0.06	Decrease	9	−1.16
	A → T	Uncharged Polar	No	0.01	Decrease	9	−1.09
14	V → A	Nonpolar	Yes	0.10	Decrease	4	−0.82
	V → L	Nonpolar	No	0.02	Decrease	6	−0.96
	V → T	Uncharged Polar	No	0.01	Decrease	10	−1.08
	V → I	Nonpolar	Yes	0.14	Decrease	8	−0.69
15	A → T	Uncharged Polar	Yes	0.16	Decrease	9	−1.20
	A → F	Nonpolar	No	0.00	Decrease	9	−0.77
	A → G	Nonpolar	No	0.02	Decrease	10	−1.59
	A → S	Uncharged Polar	No	0.03	Decrease	10	−1.28
	A → E	Acidic	Yes	0.13	Decrease	9	−1.26
	A → N	Uncharged Polar	No	0.01	Decrease	9	−1.31
17	I → A	Nonpolar	No	0.01	Decrease	4	−1.12
	I → L	Nonpolar	Yes	0.17	Decrease	1	−0.34
	I → T	Uncharged Polar	No	0.01	Decrease	8	−1.12
	I → S	Uncharged Polar	Yes	0.09	Decrease	8	−1.31
19	A → T	Uncharged Polar	Yes	0.09	Decrease	9	−1.09
	A → N	Uncharged Polar	No	0.00	Decrease	9	−1.20
	A → G	Nonpolar	No	0.01	Decrease	9	−1.56
	A → V	Nonpolar	Yes	0.15	Decrease	7	−0.54
	A → Q	Uncharged Polar	No	0.01	Decrease	9	−1.10
23	W → K	Basic	No	0.00	Decrease	9	−1.11
	W → Y	Uncharged Polar	No	0.00	Decrease	7	−0.86
	W → R	Basic	No	0.00	Decrease	7	−0.69
	W → Q	Uncharged Polar	No	0.00	Decrease	9	−1.07
25	I → V	Nonpolar	No	0.04	Decrease	7	−0.32
	I → M	Basic	No	0.01	Decrease	8	−0.81
	I → K	Basic	Yes	0.05	Decrease	9	−1.52
	I → L	Nonpolar	Yes	0.24	Decrease	5	−0.39
27	F → S	Uncharged Polar	No	0.03	Decrease	8	−1.15
	F → Y	Uncharged Polar	Yes	1.00	Decrease	2	−0.74
	F → I	Nonpolar	Yes	0.19	Decrease	5	−0.80
	F → G	Nonpolar	Yes	0.08	Decrease	7	−1.35
31	R → T	Uncharged Polar	No	0.01	Decrease	8	−0.53
	R → A	Nonpolar	No	0.01	Decrease	7	−0.49
	R → K	Basic	Yes	0.54	Decrease	9	−0.75
	R → L	Nonpolar	No	0.01	Decrease	8	−0.42
32	K → E	Acidic	Yes	0.15	Increase	2	−0.22
	K→ I	Nonpolar	No	0.00	Increase	1	−0.23
	K → R	Basic	Yes	0.15	Increase	5	0.03
	K → A	Nonpolar	No	0.01	Increase	6	−0.06

**Table 3 medsci-07-00074-t003:** Ramachandran plot statistics of blood- and brain-derived modeled Vpu structures and modeled consensus Vpu structure presenting overall model quality.

Protein Modeled	Residues in Most Favored Region	Residues in Additional Allowed Region	Residues in Generously Allowed Region	Residues in Disallowed Region	No. of Glycines	No. of Prolines
Blood-derived HIV-1 Vpu	94.7%	4.0%	1.3%	0.0%	3	1
Brain-derived HIV-1 Vpu	90.5%	8.1%	1.4%	0.0%	4	2
Consensus HIV-Vpu	92.0%	6.7%	1.3%	0.0%	3	2

**Table 4 medsci-07-00074-t004:** Presentation of binding affinity and dissociation constant (Kd) calculated from PROtein binDIng enerGY prediction (PRODIGY) and potential energy computed in SPDBV for Vpu–tetherin complexes.

Protein–Protein Complex	Model Selected	Binding Affinity ΔG (kcal/mol)	Dissociation Constant Kd (M) at 37.0 °C	Potential Energy in SPDBV (kcal/mol)
Wild-type blood-derived Vpu–tetherin	Model dock0009.pdb	−5.0	6.4 × 10^−4^	−314.17
Wild-type brain derived Vpu–tetherin	Model dock0001.pdb	−3.8	2.0 × 10^−3^	−148.65
Wild-type consensus Vpu–tetherin	Model dock0007.pdb	−4.3	9.8 × 10^−4^	−261.08

**Table 5 medsci-07-00074-t005:** This table represents the positions of mutations performed in tetherin, the best tetherin-Vpu interaction model (docked model) in Hex, binding affinity of the respective docked complexes, and their Kd values.

Name	Type of Mutation	Best Model Selected	Binding Affinity ΔG (kcal/mol)	Kd (M) at 37.0 °C	Energy (iMutant) (kcal/mol)
Tetherin	Wild-type	09	−5.0	6.4 × 10^−4^	NA
M1	L22S	10	−3.6	2.9 × 10^−3^	Decrease by −1.31
M2	L22Y	06	−3.9	3.5 × 10^−1^	Decrease by −2.06
M3	L23T	05	−2.9	3.4 × 10^−2^	Decrease by −1.52
M4	L23Y	01	−2.6	1.2 × 10^−3^	Decrease by −2.06
M5	L24F	02	−3.0	6.0 × 10^−4^	Decrease by −1.00
M6	L24M	08	−3.2	3.2 × 10^−3^	Decrease by −1.52
M7	L24T	04	−2.2	2.9 × 10^−3^	Decrease by −2.86
M8	G25A	10	−3.6	4.1 × 10^−3^	Decrease by −0.57
M9	I26S	08	−3.5	3.9 × 10^−5^	Decrease by −1.33
M10	L29Q	06	−3.9	4.4 × 10^−3^	Decrease by −0.87
M11	V30G	07	-4.1	3.9 × 10^−2^	Decrease by −2.06
M12	I33T	07	−3.5	4.2 × 10^−3^	Decrease by −1.46
M13	I34T	08	−4.3	4.5 × 10^−5^	Decrease by −1.50
M14	I36G	08	−4.1	3.5×10^−1^	Decrease by −2.05
M15	L37T	17	−4.3	3.8 × 10^−2^	Decrease by -1.86
M16	V39	14	−3.6	4.5 × 10^−1^	Decrease by −2.25
M17	P40T	11	−2.7	1.9 × 10^−3^	Decrease by −0.99
M18	L41Y	17	-3.0	3.9×10^−6^	Decrease by −1.28
M19	T45I	06	−2.5	2.7×10^−3^	Decrease by −0.77
M20	L22S, F44Y, L37I	01	−2.2	1.8 × 10^−2^	NA
M21	L23T, L37T, T45I	05	−2.9	2.3 × 10^−3^	NA

**Table 6 medsci-07-00074-t006:** This table represents the positions of mutations performed in Vpu, the best tetherin-Vpu interaction model (docked model) in Hex, binding affinity of the respective docked complexes, and their Kd values.

Name	Type of Mutation	Best Model Selected	Binding Affinity ΔG (kcal/mol)	Kd (M) at 37.0 °C	Energy (iMutant) (kcal/mol)
Vpu_blood	Wild-type	09	−5.0	6.4 × 10^−4^	NA
M1	S03Y	12	−3.6	3.3 × 10^−3^	Increase by −0.23
M2	Q05V	19	−2.9	3.4 × 10^−4^	Increase by −0.11
M3	L07S	17	−3.5	3.8 × 10^−2^	Decrease by −1.38
M4	A08T	07	−4.9	4.4 × 10^−4^	Decrease by −1.22
M5	A11F	12	−3.2	4.1 × 10^−5^	Decrease by −0.77
M6	A11L	08	−3.6	3.2 × 10^−5^	Decrease by −0.78
M7	V14K	04	−2.9	3.4 × 10^−4^	Decrease by −1.28
M8	A15L	15	−3.5	3.9 × 10^−1^	Decrease by −1.20
M9	A15T	07	−3.6	3.7 × 10^−1^	Decrease by −0.94
M10	I17A	08	−4.5	3.5 × 10^−4^	Decrease by −1.12
M11	A19T	16	−3.5	4.4 × 10^−2^	Decrease by −1.09
M12	A19H	14	−2.5	2.9 × 10^−3^	Decrease by −1.36
M13	A19F	06	−4.7	4.1 × 10^−2^	Decrease by −0.86
M14	A19L	06	−4.7	4.9 × 10^−2^	Decrease by −0.92
M15	W23K	06	−4.1	5.1 × 10^−4^	Decrease by −1.11
M16	W23L	05	−3.4	3.7 × 10^−1^	Decrease by −0.56
M17	W23Y	07	−2.1	2.1 × 10^−3^	Decrease by −0.97
M18	R31A	17	−3.3	4.8 × 10^−4^	Decrease by −0.49
M19	K32A	04	−4.2	4.9 × 10^−3^	Increase by −0.06
M20	V10K, A11L, A19T	01	−2.6	2.1 × 10^−5^	NA
M21	V14T, I18T, I26S	07	−2.7	2.2 × 10^−4^	NA
M22	A11T, V14L, A15T	02	−2.3	2.8 × 10^−2^	NA

**Table 7 medsci-07-00074-t007:** Aggregating regions predicted in tetherin by seven servers. Consensus results are highlighted as regions predicted by at least six of the seven servers.

Protein Name Accession	Position	Consensus Predicted Amyloid Regions	Fold Amyloid	Aggrescan**±	Tango**±	MetAmyl	AMYLPRED2 Consensus	Waltz **	PASTA
TetherinQ10589	**21–47** **161–176**	>sp|Q10589|BST2_HUMAN Bone marrow stromal antigen 2 OS=Homo sapiens OX=9606 GN=BST2 PE=1 SV=1 MASTSYDYCRVPMEDGDKRCKLLLGIGILVLLIIVILGVPLIIFTIKANSEACRDGLRAVMECRNVTHLLQQELTEAQKGFQDVEAQAATCNHTVMALMASLDAEKAQGQKKVEELEGEITTLNHKLQDASAEVERLRRENQVLSVRIADKKYYPSSQDSSSAAAPQLLIVLLGLSALLQ	6–11 **21–46** 58–62 93–100 144–148 **167–174**	**22–47** 92–101 **167–180**	**22–38** **168–179**	**25–49** 66–71 90–96 120–125 141–149 **160–176**	**22–47** 93–96 144–148 **167–178**	**22–47** 70–75 86–104 141–146 **166–180**	**19–50**
								Waltz± **29–47** 141–146 **166–180**	

**—Results according to Amylpred 2.0 parameters; ±—Results of individual server. Numbers in bold indicate consensus, values common in majority of servers.

**Table 8 medsci-07-00074-t008:** Propensity values predicted by Chou–Fasman secondary structure prediction program for amyloid regions in tetherin protein indicating similar propensities for sheet (E) and helix (H) formation for region 1.

Tetherin	H	E	T
Region 1: 21–47	96.3	88.9	0.0
Region 2: 161–175	75.0	37.5	12.5

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
