# Peer review of "In Silico Insights into HIV-1 Vpu-Tetherin Interactions and Its Mutational Counterparts"

_medsci, 2019, doi:10.3390/medsci7060074_

Round 1

Reviewer 1 Report

The manuscript “In silico insights into HIV-1 Vpu-tetherin interactions and its mutational counterparts” by Patil Sneha, Urmi Shah and S. Balaji describes the differential interaction of host protein Tetherin with HIV-1 Vpu derived from blood and brain. Information provided is of interest to researchers in this filed. The manuscript is well written and easy to follow. Few issues may need to be addressed. They include:

The authors should correct all grammar and spelling errors

Table 9: Please change “Chaufasman” to “Chou-Fasman”

Author Response

All the grammatical and spelling errors have been rectified in the manuscript. 

In Table 9, as well as throughout, 'chaufasman' was corrected to 'Chou-Fasman'.

Reviewer 2 Report

please see attached

Author Response

The responses to reviewer 2 is attached separately as a word document

Round 2

Reviewer 2 Report

This manuscript describes a computational study of models of complexes between HIV-1 Vpu and human

tetherin (BST-2). This version of the manuscript is a revision of a resubmission of a previous version.

In the original version, the authors built models of complexes of the transmembrane domain (TMD) α-

helices of Vpu and tetherin aligned parallel with respect to sequence direction. This is unlikely in light

of experimental work. In the resubmission, the authors have generated complex models in which the two

TMD α-helices are anti-parallel. A large portion of the results reported were labelled as H-bond interactions

between mostly hydrophobic residues, which is clearly impossible for α-helices. In this revision, the authors

have admitted this mistake and renamed these as just hydrophobic interactions.

There are no more major mistakes in the modeling results, though their value is somewhat limited. There

are still many mistakes of English grammar in the manuscript that need correction after careful proofreading.

I do not need to see such a revision.